# Deep Learning for Localization of White Matter Lesions in Neurological Diseases

Julia Machnio, Mads Nielsen, and Mostafa Mehdipour Ghazi

Pioneer Centre for Artificial Intelligence, Department of Computer Science, University of Copenhagen

## Abstract

White Matter (WM) lesions, commonly observed as hyperintensities on FLAIR MRIs or hypointensities on T1-weighted images, are associated with neurological diseases. The spatial distribution of these lesions is linked to an increased risk of developing neurological conditions, emphasizing the need for location-based analyses. Traditional manual identification and localization of WM lesions are labor-intensive and time-consuming, highlighting the need for automated solutions. In this study, we propose novel deep learning-based methods for automated WM lesion segmentation and localization. Our approach utilizes state-of-the-art models to concurrently segment WM lesions and anatomical WM regions, providing detailed insights into their distribution within the brain's anatomical structure. By applying k-means clustering to the regional WM lesion load, distinct subject groups are identified to be associated with various neurological conditions, validating the method's alignment with established clinical findings. The robustness and adaptability of our method across different scanner types and imaging protocols make it a valuable tool for research and clinical practice, offering potential improvements in diagnostic efficiency and patient care. Codes and refined atlas utilized in this study are available at https://github.com/juliamachnio/WMHLocalization.

## 1 Introduction

White matter hyperintensities (WMH) are pathological changes in white matter that typically appear as hyperintensities on FLAIR images or hypointensities on T1-weighted MRIs [1]. The total WMH load, which is the volume of WM lesion in mm³, increases with age and is recognized as an early indicator of neurological diseases such as Alzheimer's disease [2], dementia [3], and ischemic stroke [4]. Accurate diagnosis requires not only lesion detection but also quantification of their volume, shape, geometry, and location, alongside the patient's clinical symptoms [5–8]. However, manual lesion quantification performed by experienced clinicians is time-consuming and labor-intensive, underscoring the need for automated WM lesion segmentation and localization to streamline diagnosis and prognosis.

Recent studies have highlighted the critical relationship between the location of WM lesions and the future risk of developing neurological diseases, prompting a shift in research towards location-specific analysis [9]. However, these studies have primarily focused on clinical findings rather than advancing methodology, relying on time-consuming and labor-intensive approaches. For example, in [10, 11], the authors manually harmonized MRI data from thousands of patients across different cohorts to conduct voxel-based and region-of-interest analyses. While their research yielded significant insights into lesion localization, future improvements would require similarly extensive efforts, with no guarantee of consistency in results. In another study [12], the authors advanced the methodology by employing semi-automatic WM lesion detection but still relied on registration to the Montreal Neurological Institute (MNI) space for detailed statistical analysis. Despite the valuable clinical findings from these studies, the processes remained time-intensive, underscoring the need for an automated tool to streamline segmentation, registration, and WM lesion quantification.

Several studies have explored deep learning techniques for automating WM lesion segmentation and localization [13–15]. While most research has focused on WM lesion segmentation, fewer studies have addressed the detailed localization of lesions. For example, [16] introduced location information by calculating the Euclidean distance from key brain structures, such as the ventricles and cortex, to detect small WM lesions in cerebral small vessel disease. This approach was improved in [17] by integrating localization into the convolutional neural networks using multi-scale patches derived from downsampled MRIs. Another study [14] evaluated four different WM lesion segmentation algorithms and proposed registering the results to MNI space, enabling location-based assessments of WM lesion load for ischemia and lacunes. Although studies like [13, 15] have incorporated regional white matter information into automatic WM lesion segmentation and localization, a fully automated solution that bypasses the need for registration and provides practical, location-specific lesion data for both research and clinical applications is still lacking.

This study introduces deep learning-based meth-

Proceedings of the 6th Northern Lights Deep Learning Conference (NLDL), PMLR 265, 2025.

ods for WM lesion segmentation and localization, addressing the challenges of automation, speed, and localization for both clinical and research applications. Our approach focuses on segmenting WM lesions and anatomical WM regions simultaneously within the subject's native space. The key contributions of this work include: (1) developing a refined atlas of anatomical WM regions; (2) creating ground-truth labels for 170 subjects based on these WM regions; (3) training deep learning models for WM lesion segmentation and WM lesion localization; and (4) clustering the localized WM lesion loads and interpreting them with neurological diseases. This is the first method to fully automate WM region segmentation within a subject's anatomical space, eliminating the need for intra-subject or template-space registration and making it fast enough for clinical use. The obtained results are consistent with established clinical findings, differentiating distinct patient groups associated with various neurological diseases. These insights are helpful for diagnosis and prognosis, offering robust, fine-grained analyses that enhance understanding of brain changes in real-world clinical settings.

## 2 Methods

The proposed WM lesion segmentation and localization methods involve two main components, as shown in Figure 1 (B). The first component includes training deep learning networks for WM lesion segmentation, while the second focuses on WM region segmentation. Figure 1 (A) outlines the process for generating WM region ground-truth labels, which are necessary for training the deep learning models for WM region segmentation. This labelling process is performed once before model training. Ultimately, the method provides both the localization and load of WM lesions, enabling the grouping of subjects based on regional lesion similarities and facilitating connections to neurological diseases.

### 2.1 White Matter Labels

We used the JHU MNI atlas type II [19] to obtain regional WM labels for model training. We selected this atlas due to its public availability and widespread use in clinical studies related to lesion localization. The original atlas contained 130 brain regions, 24 of which were non-white matter. We merged the remaining WM labels into 34 subregions based on their ontological relationships and clinical relevance. The labels can be found in Tables A.1 and A.2.

To obtain subject-specific labels, as depicted in Figure 1 (A), we register the T1 atlas to each subject's T1 image and estimate the affine transform

from this registration. This transform is then applied to register the refined WM atlas labels to the subject's space, generating individualized labels. While using different registration methods such as SynthMorph [20] and FLIRT [21], we achieved lower accuracies than intensity-based approaches, likely due to age-related changes in WM regions. As a result, we used a multimodal intensity-based automatic image registration algorithm [22], applied exclusively to the extracted WM regions [18].

### 2.2 WM Lesion Localization

As shown in Figure 1 (B), we trained four deep learning architectures of U-Net [23], UNETR [24], MultiResUNet [25], and MedNeXt [26], to segment both WM lesions and anatomical WM regions, and combined the results to determine the location of WM lesions. We treated FLAIR and T1 images as a single modality to increase the number of training samples, improve robustness to intensity and modality variations, and enhance the model's generalizability in cases of missing data modalities. To further increase model robustness, we applied multiple augmentation techniques [27], including additive and multiplicative noise, bias field addition, rotation, elastic deformation, and motion artifact simulation. Additionally, we used a weighted combination of cross-entropy (CE) loss, Dice-Sørensen (DS) loss, and skeleton recall (SR) loss, which has proven effective for segmenting thin, tubular structures and lesions [28].

### 2.3 Inference and Clustering

The predictions from both segmentation networks were combined to localize WM lesions within each subject's anatomical space. This integration used FLAIR and T1 modalities, either separately or in combination, by aggregating probability scores before applying hard pixel classification. Next, we applied $k$-means clustering to the normalized regional lesion loads extracted from the labeled WM lesions. This allowed us to group patients based on regional similarities in WM lesion distributions and to associate these groups with brain-related diseases reported in the literature.

## 3 Experiments and Results

### 3.1 Data

We conducted all experiments using the publicly available MICCAI 2017 WMH Segmentation Challenge dataset [29], which includes 3D T1 and FLAIR images from 170 subjects across three distinct cohorts: Utrecht, Amsterdam, and Singapore. The

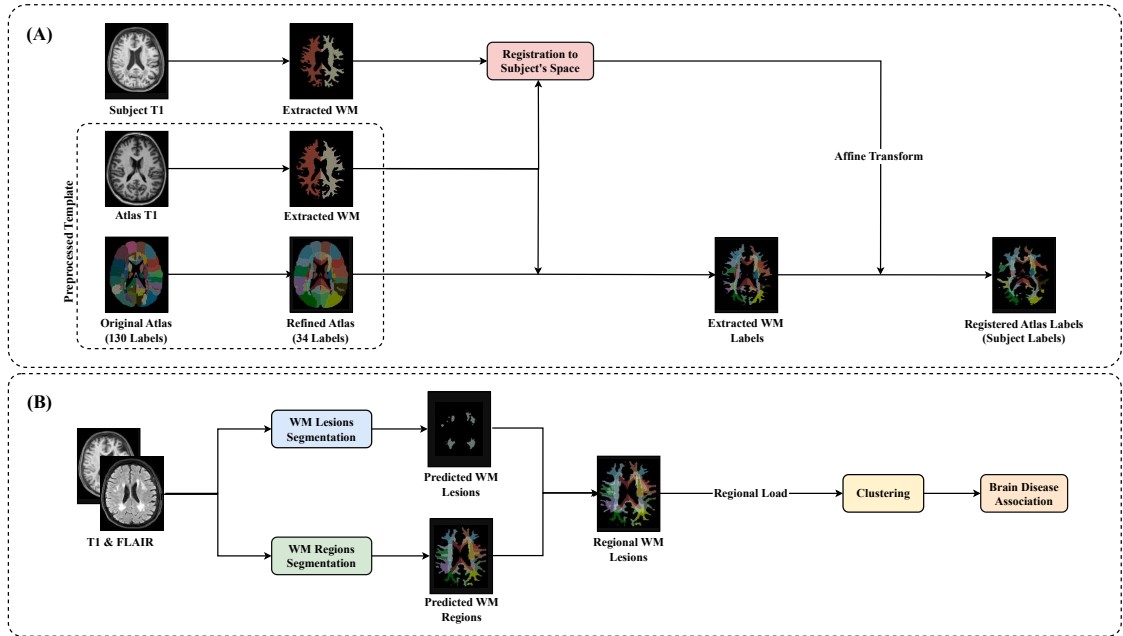

**Figure 1.** Overview of the proposed methods for WM lesion and region segmentation. (A) The process for creating ground-truth WM region labels for individual scans. First, WM regions are segmented [18] in the subject and the atlas. The atlas WM is then registered to the subject's WM using affine transforms, and these transforms are applied to map the atlas labels into the subject's anatomical space. (B) Deep learning pipeline for regional WM lesion segmentation and analysis. Two deep networks are trained on T1 and FLAIR images for segmenting WM lesions and for segmenting WM regions. The predictions from these models are combined to provide regional WM lesions for each subject. The regional lesion loads are then calculated and used for clustering to explore associations with various neurological conditions.

**Table 1.** Overview of the dataset splits used in this study. The vendor abbreviations refer to GE (G), Philips (P), and Siemens (S).

| Dataset | #Scans | Dimensions | Resolution | Strength | Vendor |
|---|---|---|---|---|---|
| **Train** | 110 | 181×251×81 | 1.1×1×3mm$^3$ | 1.5T,3T | G, P, S |
| **Test** | 60 | 202×250×60 | 1.1×0.98×3mm$^3$ | 3T | G, P, S |

dataset provides manually annotated labels for background, WM lesions, and other pathologies. The data was divided into training and testing sets, as detailed in Table 1, by swapping the challenge-provided training and testing splits to increase the amount of training data for WMH localization. For training, the subjects were randomly split into training and validation subsets using 5-fold cross-validation, with an 80:20 ratio.

## 3.2 Settings

To select the best-performing effective models, we trained both 2D and 3D architectures, including U-Net [23], UNETR [24], MultiResUNet [25], and MedNeXt [26], for WM lesion and region segmentation using the training splits from all three cohorts. This process was also repeated for each cohort individually. Each model was evaluated using T1, FLAIR, and an ensemble of both modalities based

on Softmax outputs. The most accurate models were then selected to calculate regional lesion loads for scan clustering.

The models were primarily trained using a weighted combination of the CE and DS loss functions. This process was then repeated with the addition of the SR loss. The losses were weighted equally since their values were observed to be within a similar range during training. All models were implemented in a PyTorch framework [30] and optimized using the Stochastic Gradient Descent (SGD) method with Nesterov momentum set to 0.9 and an initial learning rate of 0.001. Training was conducted for 1,000 iterations with a maximum of 250 mini-batches.

## 3.3 Results

We first applied the trained WM lesion segmentation models to the test data. Models evaluated using only FLAIR images achieved significantly higher accuracy than those evaluated with T1 or combined modalities (with a $p$-value $\ll 0.05$ using the Wilcoxon signed-rank test). This is likely due to the clearer distinction between WM lesion and WM tissue intensities in FLAIR images compared to T1. Additionally, models trained with the combined loss function, including SR, consistently achieved supe-

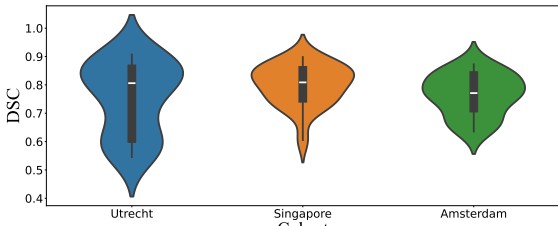

**Figure 2.** Violin plots depicting test WM lesion segmentation accuracy across different cohorts. The results were obtained using a 2D U-Net model trained with the combination of CE, DS, and SR loss functions.

**Table 2.** Test segmentation accuracy (mean ± SD) for WM lesion and WM region segmentation tasks using T1, FLAIR, and ensemble modalities. For WM lesion segmentation, results are from a 2D U-Net model trained with CE, DS, and SR loss functions. For WM region segmentation, results are from a 3D MultiResUNet model trained with CE and DS loss functions. Statistically significant differences are highlighted for each task ($p$-value $\ll 0.05$ using the Wilcoxon signed-rank test).

| Segmentation Task | T1 | FLAIR | T1 & FLAIR |
|---|---|---|---|
| WM lesions | $0.59 \pm 0.17$ | $\mathbf{0.77 \pm 0.09}$ | $0.73 \pm 0.11$ |
| WM regions | $\mathbf{0.82 \pm 0.04}$ | $0.75 \pm 0.03$ | $0.80 \pm 0.03$ |

rior accuracy. The SR loss utilizes additional masks that preserve structural connectivity, improving the segmentation of small and thin structures like WM lesions. Detailed WM lesion segmentation results are presented in Tables B.1 to B.5.

The 2D U-Net model trained with the CE, DS, and SR loss functions and evaluated solely on FLAIR images achieved the highest segmentation accuracy, with a Dice similarity coefficient (DSC) of 0.77, comparable to current state-of-the-art results [29, 31, 32]. Accuracy variations may arise from differences in data splits and augmentation techniques we employed to develop a more robust model. Additionally, we generated violin plots to illustrate WM lesion segmentation accuracy across different cohorts using the best-performing model (see Figure 2). The Singapore and Amsterdam cohorts displayed similar dispersion and interquartile ranges, while the Utrecht cohort exhibited higher dispersion, likely due to its varying data resolution.

Next, we applied the trained WM region segmentation models to the test data. The 3D MultiResUNet model, trained with the CE and DS loss functions, achieved the highest DSC, as shown in the violin plots in Figure 3 using T1 images. These plots demonstrate similar dispersion for the left and right WM compartments. Differences in dispersion between regions strongly depend on the region's size and shape. For example, region 11, with an average DSC of 0.68, consists of three small gyrus parts with high curvature that are not always connected. Conversely, region 12 has a large region with fewer

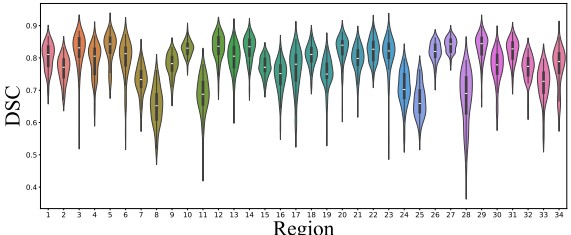

**Figure 3.** Violin plots of test WM regions segmentation accuracy across all cohorts. Results were obtained using a 3D MultiResUNet model trained with CE and DS loss functions. The plots illustrate that larger regions with simpler curvatures, such as regions 10 (superior, inferior, and middle temporal gyrus) and 12 (superior corona radiata and superior longitudinal fasciculus), tend to have higher accuracy. Detailed descriptions of each region are provided in Tables A.1 and A.2.

curvatures including superior corona radiata and superior longitudinal fasciculus, resulting in an average DSC of 0.82. Models evaluated on T1 images showed significantly higher segmentation accuracy (with a $p$-value $\ll 0.05$ using the Wilcoxon signed-rank test). Table 2 summarizes the best results for the segmentation tasks used in the later analysis.

Unlike WM lesion segmentation, T1 images provided better contrast between tissues and a higher signal-to-noise ratio, leading to a more accurate anatomical WM segmentation. There was a notable accuracy difference between 2D and 3D models, as the varying morphology of WM regions is better preserved in 3D. No significant accuracy differences were observed between models trained with the CE and DS and those with the SR loss functions, except for training time where models trained with SR converged faster. All WM region segmentation results are detailed in Tables C.1 to C.5.

We further localized and evaluated the regional WM lesions by multiplying the predicted WM lesion masks with the regional WM labels. Figure 4 displays a coronal view of the predicted regional WM lesion labels with the highest lesion loads across test subjects. As can be seen, WM lesions are nearly symmetric on both sides of the brain, particularly in the superior-anterior part of the corona radiata (CR) and the limb of the internal capsule. However, in region 30, which primarily comprises the posterior part of the CR, most WM lesions appeared on the right side. In contrast, in region 7 (cuneus, precuneus, and lingual gyrus), WM lesions were found exclusively on the left side.

Finally, we calculated the regional WMH loads and performed $k$-means clustering on them. Figure 5 and Table D.1 present the results of this experiment, where we achieved a silhouette index of 0.51 for $K = 4$ groups. Subjects were grouped based on the distribution of WM lesions across different regions. Group 1, which exhibited the lowest WM lesion

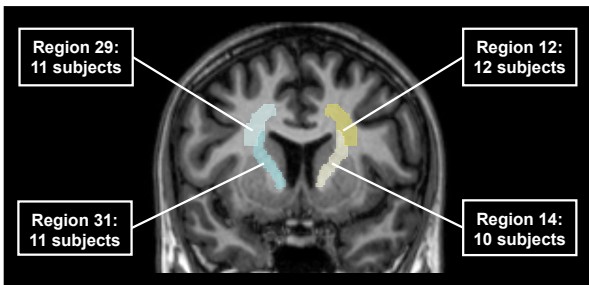

**Figure 4.** Occurrence of predicted regional WM lesions with the highest load across test subjects. WM lesions are almost symmetrical on both sides of the brain.

load, had lesions located primarily in the anterior regions on both sides of the brain. As the regional WM lesion load increased, lesions shifted to superior regions, particularly in the left hemisphere for those with the highest lesion load. Besides, the clustering algorithm identified two subgroups with medium WM lesion loads (groups 2 and 4). In group 2, most lesions were concentrated in superior regions, similar to group 3. However, in group 4, WM lesions predominantly appeared in the posterior regions of the brain, as shown in the regional WM lesion maps.

## 3.4 Discussion

The WM lesion scatter analysis revealed that for smaller regional lesion loads, lesions predominantly appeared in the CR, the anterior limb of the internal capsule, and the superior fronto-occipital fasciculus on both sides of the brain. As the WM lesion volume increased, the lesions shifted to superior regions, including the superior longitudinal fasciculus and superior CR on both hemispheres. Notably, for regional WM lesion loads above 45 ml, the majority of lesions were observed on the left side of the brain. Previous studies, such as [33], have reported a direct correlation between WM lesions in subcortical white matter regions (groups 2 and 3) and a heightened risk of ischemic stroke. These regions are critical for connecting various brain areas and facilitating information transfer. Hence, WM lesions in these areas can disrupt cognitive processes, potentially indicating future cognitive decline [6] and the onset of Alzheimer's disease [34].

Additionally, we identified a subgroup with medium WM lesion loads, where most lesions were located in the posterior thalamic radiation, posterior corona radiata, tapetum, and retrolenticular part of the internal capsule, predominantly on the right side. The literature study in [35] highlighted that the accumulation of WM lesions, especially in the anteroposterior regions, significantly contributes to cerebral amyloid angiopathy, which is associated with brain hemorrhages.

Furthermore, a few subjects showed a predomi-

nant occurrence of WM lesions in the lingual gyrus, precuneus, and cuneus within the left hemisphere. These regions are essential for language processing and memory [7], as the left hemisphere is primarily responsible for analytical processing, reading, and writing. They also play a crucial role in verbal and narrative memory retrieval. Consequently, WM lesions in these areas can disrupt verbal fluency and language processing, potentially indicating conditions such as multiple sclerosis or stroke [7].

## 4 Conclusion

In this study, we introduced novel methods for the fully automated segmentation and localization of white matter lesions within the brain. Our approach offers deep learning-based operations directly within the subject's anatomical space, addressing a significant clinical research need for a robust and efficient tool suitable for daily use. We validated the method's compliance with recent clinical findings and highlighted its importance in providing critical insights for predicting neurological diseases.

The method demonstrated effectiveness across various scanner types and protocols, showcasing its versatility and robustness. By leveraging advanced deep learning techniques, including the integration of FLAIR and T1 modalities, MRI-based data augmentation, and a combination of state-of-the-art loss functions, we achieved high accuracy in WM lesion segmentation and localization. Our results underscore the method's potential to enhance research and clinical workflows, ultimately improving the management of neurological conditions.

## Acknowledgments

This project has received funding from Innovation Fund Denmark under grant number 1063-00014B, Lundbeck Foundation with reference number R400-2022-617, and Pioneer Centre for AI, Danish National Research Foundation, grant number P1.

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

# A    Anatomical WM Labels

# B    Supplementary WM lesion segmentation results

# C    Supplementary WM region segmentation results

# D    Supplementary WM lesion clustering results

**Table A.1.** The refined JHU MNI type II atlas regions for the left hemisphere. The refined labels include only regions within the brain's white matter. The Merged Label column lists newly created regional labels, while the Atlas Label and Region columns provide the original atlas labels and their corresponding anatomical names.

| WM | Merged label | Atlas label | Region |
|---|---|---|---|
| Left | 1 | 1 | Superior Parietal Lobule |
| | | 8 | Angular Gyrus |
| | | 23 | Supramarginal Gyrus |
| | 2 | 2 | Cingulate Gyrus |
| | | 40 | Cingulum (Cingulate Gyrus) |
| | 3 | 3 | Superior Frontal Gyrus |
| | 4 | 4 | Middle Frontal Gyrus |
| | 5 | 5 | Inferior Frontal Gyrus |
| | 6 | 6 | Precentral Gyrus |
| | | 7 | Postcentral Gyrus |
| | 7 | 9 | Pre-cuneus |
| | | 10 | Cuneus |
| | | 11 | Lingual Gyrus |
| | 8 | 12 | Fusiform Gyrus |
| | | 13 | Parahippocampal Gyrus |
| | | 17 | Entorhinal Area |
| | | 41 | Cingulum (hippocampus) |
| | 9 | 14 | Superior Occipital Gyrus |
| | | 15 | Inferior Occipital Gyrus |
| | | 16 | Middle Occipital Gyrus |
| | 10 | 18 | Superior Temporal Gyrus |
| | | 19 | Inferior Temporal Gyrus |
| | | 20 | Middle Temporal Gyrus |
| | 11 | 21 | Lateral Fronto-Orbital Gyrus |
| | | 22 | Middle Front-Orbital Gyrus |
| | | 24 | Gyrus Rectus |
| | 12 | 38 | Superior Corona Radiata |
| | | 43 | Superior Longitudinal Fasciculus |
| | 13 | 36 | Posterior Thalamic Radiation |
| | | 39 | Posterior Corona Radiata |
| | | 55 | Retrolenticular Part Of Internal Capsule |
| | | 58 | Tapetum |
| | 14 | 34 | Anterior Limb Of Internal Capsule |
| | | 37 | Anterior Corona Radiata |
| | | 44 | Superior Fronto-Occipital Fasciculus |
| | 15 | 25 | Insular |
| | | 45 | Inferior Fronto-Occipital Fasciculus |
| | | 47 | External Capsule |
| | | 48 | Uncinate Fasciculus |
| | 16 | 33 | Cerebral peduncle |
| | | 35 | Posterior Limb Of Internal Capsule |
| | | 42 | Fornix(cres) Stria Terminalis |
| | | 46 | Saggital Stratum |
| | 17 | 51 | Fornix (Column And Body) |
| | | 52 | Genu Of Corpus Callosum |
| | | 53 | Body Of Corpus Callosum |
| | | 54 | Selenium Of Corpus Callosum |

**Table A.2.** The refined JHU MNI type II atlas regions for the right hemisphere. The refined labels include only regions within the brain's white matter. The Merged Label column lists newly created regional labels, while the Atlas Label and Region columns provide the original atlas labels and their corresponding anatomical names.

| WM | Merged label | Atlas label | Region |
|---|---|---|---|
| Right | 18 | 66 | Superior Parietal Lobule |
| | | 73 | Angular Gyrus |
| | | 88 | Supramarginal Gyrus |
| | 19 | 67 | Cingulate Gyrus |
| | | 105 | Cingulum (Cingulate Gyrus) |
| | 20 | 68 | Superior Frontal Gyrus |
| | 21 | 69 | Middle Frontal Gyrus |
| | 22 | 70 | Inferior Frontal Gyrus |
| | 23 | 71 | Precentral Gyrus |
| | | 72 | Postcentral Gyrus |
| | 24 | 74 | Pre-cuneus |
| | | 75 | Cuneus |
| | | 76 | Lingual Gyrus |
| | 25 | 77 | Fusiform Gyrus |
| | | 78 | Parahippocampal Gyrus |
| | | 82 | Entorhinal Area |
| | | 106 | Cingulum (hippocampus) |
| | 26 | 79 | Superior Occipital Gyrus |
| | | 80 | Inferior Occipital Gyrus |
| | | 81 | Middle Occipital Gyrus |
| | 27 | 83 | Superior Temporal Gyrus |
| | | 84 | Inferior Temporal Gyrus |
| | | 85 | Middle Temporal Gyrus |
| | 28 | 86 | Lateral Fronto-Orbital Gyrus |
| | | 87 | Middle Front-Orbital Gyrus |
| | | 89 | Gyrus Rectus |
| | 29 | 103 | Superior Corona Radiata |
| | | 108 | Superior Longitudinal Fasciculus |
| | 30 | 101 | Posterior Thalamic Radiation |
| | | 104 | Posterior Corona Radiata |
| | | 120 | Retrolenticular Part Of Internal Capsule |
| | | 123 | Tapatum |
| | 31 | 99 | Anterior Limb Of Internal Capsule |
| | | 102 | Anterior Corona Radiata |
| | | 109 | Superior Fronto-Occipital Fasciculus |
| | 32 | 90 | Insular |
| | | 110 | Inferior Fronto-Occipital Fasciculus |
| | | 112 | External Capsule |
| | | 113 | Uncinate Fasciculus |
| | 33 | 98 | Cerebral peduncle |
| | | 100 | Posterior Limb Of Internal Capsule |
| | | 107 | Fornix(cres) Stria Terminalis |
| | | 111 | Saggital Stratum |
| | 34 | 116 | Fornix (Column And Body) |
| | | 117 | Genu Of Corpus Callosum |
| | | 118 | Body Of Corpus Callosum |
| | | 119 | Selenium Of Corpus Callosum |

**Table B.1.** Test WM lesion segmentation accuracy (mean ± SD) for various architectures and modalities applied to all cohorts. The models were trained with CE and DS loss functions. Statistically significant differences between modalities for models that achieved the highest accuracy are highlighted in bold ($p$-value $\ll$ using the Wilcoxon signed-rank test).

| Model | | ALL | | |
| --- | --- | --- | --- | --- |
| | | T1 | FLAIR | T1 & FLAIR |
| U-Net | 2D | 0.57 ± 0.16 | **0.75 ± 0.10** | 0.69 ± 0.13 |
| | 3D | 0.59 ± 0.16 | **0.75 ± 0.11** | 0.67 ± 0.14 |
| U-NetR | 2D | 0.43 ± 0.16 | 0.59 ± 0.16 | 0.56 ± 0.17 |
| | 3D | 0.45 ± 0.15 | 0.67 ± 0.13 | 0.61 ± 0.14 |
| MedNeXt | 2D | 0.49 ± 0.16 | 0.57 ± 0.21 | 0.54 ± 0.18 |
| | 3D | 0.53 ± 0.16 | 0.68 ± 0.15 | 0.61 ± 0.16 |
| MultiResUNet | 2D | 0.45 ± 0.18 | 0.42 ± 0.21 | 0.46 ± 0.19 |
| | 3D | 0.61 ± 0.16 | 0.73 ± 0.11 | 0.68 ± 0.14 |

**Table B.2.** Test WM lesion segmentation accuracy (mean ± SD) for various architectures and modalities applied to the Utrecht cohort. The models were trained with CE and DS loss functions. Statistically significant differences between modalities for a model that achieved the highest accuracy are highlighted in bold ($p$-value $\ll$ using the Wilcoxon signed-rank test).

| Model | | Utrecht | | |
| --- | --- | --- | --- | --- |
| | | T1 | FLAIR | T1 & FLAIR |
| U-Net | 2D | 0.57 ±0.17 | **0.74 ± 0.14** | 0.68± 0.16 |
| | 3D | 0.61 ± 0.16 | 0.72 ± 0.15 | 0.72 ± 0.15 |
| U-NetR | 2D | 0.49 ± 0.16 | 0.68 ± 0.15 | 0.64 ± 0.13 |
| | 3D | 0.46 ± 0.14 | 0.63 ± 0.18 | 0.60 ± 0.14 |
| MedNeXt | 2D | 0.53 ± 0.16 | 0.69 ± 0.15 | 0.64 ± 0.14 |
| | 3D | 0.53 ± 0.17 | 0.66 ± 0.15 | 0.61 ± 0.16 |
| MultiResUNet | 2D | 0.50 ± 0.19 | 0.55 ± 0.21 | 0.46 ± 0.19 |
| | 3D | 0.61 ± 0.18 | 0.72 ± 0.14 | 0.68 ± 0.16 |

**Table B.3.** Test WM lesion segmentation accuracy (mean ± SD) for various architectures and modalities applied to the Singapore cohort. The models were trained with CE and DS loss functions. Statistically significant differences between modalities for a model that achieved the highest accuracy are highlighted in bold ($p$-value $\ll$ using the Wilcoxon signed-rank test).

| Model | | Singapore | | |
| --- | --- | --- | --- | --- |
| | | T1 | FLAIR | T1 & FLAIR |
| U-Net | 2D | 0.64 ± 0.14 | 0.76 ± 0.08 | 0.70 ±0.11 |
| | 3D | 0.65 ± 0.13 | **0.77 ± 0.10** | 0.71 ± 0.11 |
| U-NetR | 2D | 0.40 ± 0.19 | 0.54 ± 0.13 | 0.42 ± 0.17 |
| | 3D | 0.51 ± 0.19 | 0.73 ± 0.10 | 0.62 ± 0.16 |
| MedNeXt | 2D | 0.49 ± 0.18 | 0.35 ± 0.12 | 0.35 ± 0.14 |
| | 3D | 0.58 ± 0.16 | 0.68 ± 0.12 | 0.62± 0.15 |
| MultiResUNet | 2D | 0.49 ± 0.17 | 0.38 ±0.17 | 0.46 ± 0.18 |
| | 3D | 0.68 ± 0.14 | 0.73 ± 0.10 | 0.71±0.13 |

**Table B.4.** Test WM lesion segmentation accuracy (mean ± SD) for various architectures and modalities applied to the Amsterdam cohort. The models were trained with CE and DS loss functions. Statistically significant differences between modalities for a model that achieved the highest accuracy are highlighted in bold ($p$-value $\ll$ using the Wilcoxon signed-rank test).

| Model | | Amsterdam | | |
| --- | --- | --- | --- | --- |
| | | T1 | FLAIR | T1 & FLAIR |
| U-Net | 2D | 0.51 ± 0.15 | 0.74 ± 0.14 | 0.67 ± 0.10 |
| | 3D | 0.54 ± 0.14 | **0.75 ± 0.09** | 0.65 ± 0.12 |
| U-NetR | 2D | 0.41 ± 0.13 | 0.57 ± 0.16 | 0.60 ± 0.12 |
| | 3D | 0.39 ± 0.11 | 0.67 ± 0.11 | 0.59 ± 0.11 |
| MedNeXt | 2D | 0.46 ± 0.13 | 0.69 ± 0.11 | 0.63 ± 0.12 |
| | 3D | 0.48 ± 0.13 | 0.70 ± 0.10 | 0.61 ± 0.11 |
| MultiResUNet | 2D | 0.35 ± 0.16 | 0.34 ± 0.19 | 0.38 ± 0.16 |
| | 3D | 0.56 ± 0.14 | 0.74 ± 0.09 | 0.64 ± 0.14 |

**Table B.5.** Test WM lesion segmentation accuracy (mean ± SD) for various architectures and modalities applied to all cohorts. The models were trained with CE, DS, and SR loss functions. Statistically significant differences between modalities for a model that achieved the highest accuracy are highlighted in bold ($p$-value $\ll$ using the Wilcoxon signed-rank test).

| Model | | All | | |
| --- | --- | --- | --- | --- |
| | | T1 | FLAIR | T1 & FLAIR |
| U-Net | 2D | 0.59 ± 0.17 | **0.77 ± 0.09** | 0.73 ± 0.11 |
| | 3D | 0.59 ± 0.16 | 0.72 ± 0.12 | 0.71 ± 0.12 |
| U-NetR | 3D | 0.47 ± 0.20 | 0.62 ± 0.19 | 0.62 ± 0.16 |
| MedNeXt | 3D | 0.75 ± 0.05 | 0.61 ± 0.11 | 0.70 ± 0.06 |
| MultiResUNet | 3D | 0.59 ± 0.17 | 0.72 ± 0.13 | 0.71 ± 0.12 |

**Table C.1.** Test WM region segmentation accuracy (mean ± SD) for various architectures and modalities applied to all cohorts. The models were trained with CE and DS loss functions. Statistically significant differences between modalities for a model that achieved the highest accuracy are highlighted in bold ($p$-value $\ll$ using the Wilcoxon signed-rank test).

| Model | | All | | |
| --- | --- | --- | --- | --- |
| | | T1 | FLAIR | T1 & FLAIR |
| U-Net | 2D | 0.75 ± 0.01 | 0.70 ±0.01 | 0.74 ±0.01 |
| | 3D | 0.80 ± 0.03 | 0.77 ± 0.04 | 0.80 ± 0.04 |
| U-NetR | 2D | 0.61 ± 0.10 | 0.40 ±0.26 | 0.50 ±0.23 |
| | 3D | 0.74 ±0.05 | 0.69 ±0.05 | 0.74 ± 0.05 |
| MedNeXt | 2D | 0.75 ± 0.05 | 0.61 ± 0.11 | 0.70 ± 0.06 |
| | 3D | 0.80± 0.03 | 0.74 ± 0.05 | 0.78 ± 0.04 |
| MultiResUNet | 2D | 0.71±0.04 | 0.65 ±0.05 | 0.70 ±0.05 |
| | 3D | **0.81 ±0.04** | 0.75 ±0.04 | 0.79 ± 0.03 |

**Table C.2.** Test WM region segmentation accuracy (mean ± SD) for various architectures and modalities applied to the Utrecht cohort. The models were trained with CE and DS loss functions. Statistically significant differences between modalities for a model that achieved the highest accuracy are highlighted in bold ($p$-value $\ll$ using the Wilcoxon signed-rank test).

| Model | | Utrecht | | |
| --- | --- | --- | --- | --- |
| | | T1 | FLAIR | T1 & FLAIR |
| U-Net | 2D | 0.74 ± 0.03 | 0.69 ±0.02 | 0.74 ± 0.02 |
| | 3D | **0.79 ±0.04** | 0.74 ± 0.05 | 0.78 ±0.04 |
| U-NetR | 2D | 0.70 ±0.1 | 0.62 ±0.11 | 0.69 ±0.10 |
| | 3D | 0.69 ± 0.04 | 0.60 ± 0.07 | 0.69 ± 0.04 |
| MedNeXt | 2D | 0.73 ± 0.11 | 0.67 ± 0. 11 | 0.72 ± 0.12 |
| | 3D | 0.76 ± 0.04 | 0.69 ± 0.06 | 0.75 ± 0.04 |
| MultiResUNet | 2D | 0.72 ±0.08 | 0.53 ± 0.08 | 0.67 ± 0.08 |
| | 3D | **0.79 ± 0.03** | 0.74 ± 0.04 | 0.78 ± 0.03 |

**Table C.3.** Test WM region segmentation accuracy (mean ± SD) for various architectures and modalities applied to the Singapore cohort. The models were trained with CE and DS loss functions. Statistically significant differences between modalities for models that achieved the highest accuracy are highlighted in bold ($p$-value $\ll$ using the Wilcoxon signed-rank test).

| Model | | Singapore | | |
| | | T1 | FLAIR | T1 & FLAIR |
|---|---|---|---|---|
| U-Net | 2D | 0.74 ±0.01 | 0.70 ±0.01 | 0.74 ± 0.01 |
| | 3D | **0.77 ±0.05** | 0.74 ±0.06 | **0.77 ±0.05** |
| U-NetR | 2D | 0.63 ±0.08 | 0.54 ± 0.17 | 0.61 ±0.16 |
| | 3D | 0.59 ± 0.09 | 0.63 ±0.07 | 0.65 ± 0.08 |
| MedNeXt | 2D | 0.73 ± 0.04 | 0.69 ± 0.05 | 0.73 ± 0.04 |
| | 3D | 0.73 ± 0.05 | 0.70 ± 0.06 | 0.73 ± 0.05 |
| MultiResUNet | 2D | 0.65 ± 0.05 | 0.69 ±0.05 | 0.71 ± 0.05 |
| | 3D | 0.74 ± 0.05 | 0.74 ± 0.06 | 0.74 ± 0.05 |

**Table C.4.** Test WM region segmentation accuracy (mean ± SD) for various architectures and modalities applied to the Amsterdam cohort. The models were trained with CE and DS loss functions. Statistically significant differences between modalities for models that achieved the highest accuracy are highlighted in bold ($p$-value $\ll$ using the Wilcoxon signed-rank test).

| Model | | Amsterdam | | |
| | | T1 | FLAIR | T1 & FLAIR |
|---|---|---|---|---|
| U-Net | 2D | 0.55 ± 0.03 | 0.54 ±0.04 | 0.57 ± 0.03 |
| | 3D | **0.81 ± 0.03** | 0.76 ± 0.04 | 0.79 ± 0.04 |
| U-NetR | 2D | 0.34 ±0.09 | 0.29 ± 0.09 | 0.32 ±0.10 |
| | 3D | 0.75 ± 0.05 | 0.70 ±0.04 | 0.75 ±0.04 |
| MedNeXt | 2D | 0.49 ± 0.20 | 0.50 ± 0.12 | 0.53 ±0.16 |
| | 3D | 0.80 ± 0.03 | 0.75 ± 0.03 | 0.78 ± 0.03 |
| MultiResUNet | 2D | 0.42 ± 0.19 | 0.31 ± 0.16 | 0.37 ± 0.20 |
| | 3D | **0.81 ± 0.03** | 0.77 ± 0.04 | 0.80 ± 0.03 |

**Table C.5.** Test WM region segmentation accuracy (mean ± SD) for various architectures and modalities applied to all cohorts. The models were trained with CE, DS, and SR loss functions. Statistically significant differences between modalities for a model that achieved the highest accuracy are highlighted in bold ($p$-value $\ll$ using the Wilcoxon signed-rank test).

| Model | | All | | |
| | | T1 | FLAIR | T1 & FLAIR |
|---|---|---|---|---|
| U-Net | 3D | 0.80 ± 0.03 | 0.76 ± 0.05 | 0.79 ± 0.04 |
| U-NetR | 3D | 0.74 ± 0.06 | 0.69 ± 0.05 | 0.73 ± 0.04 |
| MedNeXt | 3D | 0.78 ± 0.04 | 0.74 ± 0.04 | 0.77 ± 0.03 |
| MultiResUNet | 3D | **0.81 ± 0.04** | 0.76 ± 0.04 | 0.80 ± 0.04 |

**Table D.1.** Details of the groups obtained by $k$-means clustering of the regional WM lesion loads.

| | #Subjects | Mean WM lesions load | Min WM lesions load | Max WM lesions load | Major WM lesions region(s) | #Subjects WM lesions left side | #Subjects WM lesions right side |
|---|---|---|---|---|---|---|---|
| **Group 1** | 29 | 629 | 57 | 2457 | 14, 31 | 14 | 15 |
| **Group 2** | 15 | 5562 | 2338 | 8651 | 29, 12 | 8 | 7 |
| **Group 3** | 7 | 14121 | 8730 | 20159 | 12, 29 | 6 | 1 |
| **Group 4** | 9 | 4853 | 3388 | 8172 | 30 | 4 | 5 |

