# OpenReview forum: "Deep Learning for Localization of White Matter Lesions in Neurological Diseases"
_NLDL.org/2025/Conference — NLDL 2025 Oral_

### Official Review · Reviewer_kk7q · 2024-10-06
**Review for "Deep Learning for Localization of White Matter Lesions in Neurological Diseases"**

**Confidence:** 4

**Summary:**

The authors present a pipeline for segmenting white matter lesions and regions. Their findings indicate that grouping patients based on their detected regional white matter hyperintensity load relates to clinical conditions.

**Strengths:**

1) The paper is well-written, without apparent errors.
2) Clustering patients based on their regional white matter hyperintensity load is an interesting approach, though further clarification could enhance understanding.
3) The use of publicly available data (MICCAI 2017 WMH Segmentation Challenge) adds transparency and reproducibility to the research.
4) The authors provide comprehensive experimental results using various architectures and inputs. The Appendix section is particularly valuable.

**Weaknesses:**

1) The authors do not share their code, which makes it very difficult to reproduce or understand some parts of the paper.
2) Clarity and presentation could be further improved:
   * Figure 1 is not very informative and it should be the main figure of the paper. Apart from reformating it, minor things like increasing the fonts could help.
   * Contributions (1) and (2) are closely related but abruptly mentioned without the previous introduction of the underlying problem. Additionally, contribution (1) would rely upon the subsequent sharing of the "atlas".
   * Contribution (3), "training" deep learning models cannot be considered a contribution by itself. Throughout the paper, the novelty does not lie in the models used, as they are state-of-the-art architectures. The paper would benefit from a clearer emphasis on its unique contributions, which can be argued to exist but are not highlighted effectively.
3) The limitations of the paper are not discussed.
4) In the Introduction, the previous and related work should be clearer, which can be exemplified by the following questions:
    * Is this "the first method to fully automate WM region segmentation within a subject's anatomical space" (line 103)? Or is it the localization that is done for the first time in the native space?
   * What are the disadvantages of working in the MNI space? Speed? Is the potential application of this method speed-dependent?

**Final Rebuttal Confidence:**

4

**Final Rebuttal Justification:**

Thank you to the authors for their thorough revision of the paper. Here are my key points:

* The public availability of the code is essential for this kind of publication. Additionally, sharing their atlas would enhance their contributions (claim 2).
* The process for generating the ground truth has been better articulated, and I hope the revised Figure 1 further supports this aspect.
* Detailed explanations of the background for the contributions are crucial for understanding the paper's significance. I look forward to these in the revised version.
* I appreciate the addition of a limitations section in the discussion, especially considering the other reviewers' comments.

Overall, if the authors implement the changes mentioned in their rebuttal, I would be willing to raise my initial rating.

**Justification:**

The paper does not have any noticeable writing errors and the appendix section is actually helpful. The experimentation on a public dataset and the idea of grouping the patients by their regional WMH load are noteworthy as well.

However, the paper lacks clarity in key areas. The reader struggles to know whether the authors present novel methods for segmentation, apply available methods in novel ways, or neither. Sentences such as "This study introduces deep learning-based methods" (line 089) are vague and contribute to this misunderstanding. This should not be a problem as long as this distinction is clear. Sharing the code would also address some misunderstandings and further improve reproducibility.

Additionally, clarifying the contributions and reformatting Figure 1 to better communicate the main information would address the bigger issues. Better framing of the problem in the Introduction and a discussion of the study’s limitations would further aid readers in following the paper’s structure.

Overall, while these issues are fixable, given the current state of the paper, my recommendation is borderline rejection.

---

> ### Author Rebuttal · Authors · 2024-10-22
>
> Dear receiver kk7q,
>
> Thank you for notifying the strengths and weaknesses of our paper. We adress the mentioned issues/questions below.
>
>
> 1.	Public availability of the code
>
> We fully acknowledge the importance of it and are committed to ensuring the public availability of our code. We believe that this transparency not only aids in the reproducibility of the research but also enhances the understanding of unclear points. As a result, we will share the code in a publicly available repository on github, along with a refined atlas in MNI space in final version of the paper.
>
> 2.	Figure 1 is not very informative and it should be the main figure of the paper. Apart from reformating it, minor things like increasing the fonts could help
>
> We updated image 1A to dispel doubts about the process and introduced mentioned visual suggestions such as font size.
>
> The process of generating GT labels for region segmentation consists of the following steps:
>
> 1.	The white matter is extracted separately from subject T1 data and atlas T1.
>
> 2. Then the registration of the extracted atlas's WM to the subject's WM is performed to obtain affine transformation (Update 1 to the image: The "registration" block is moved up to the level of "subject T1" and renamed as "Registration to subject’s space" to indicate to which space we are registering.
>
> 3. The parallel process (done only once) is refining the atlas. The original atlas contains 130 labels and is too detailed; thus, it is refined into 34 regions, similar to other existing studies. (Update 2 to the image: We added the 'original atlas" image below "Atlas T1" that is pointing into Atlas labels refined, also moved to the bottom of block 1A.)
>
> 4. The last step is to apply the transformation obtained in step 2 to the 'atlas labels refined", which transforms them into subject space, providing subjects' ground truth labels. (Update 3 to the image: blocks from points 2 and 3 are combined, pointing out the final result - "Registered atlas labels").
>
> We hope the process, especially the image, will be straightforward and understandable.
>
> 3.	Contributions (1) and (2) are closely related but abruptly mentioned without the previous introduction of the underlying problem. Additionally, contribution (1) would rely upon the subsequent sharing of the "atlas".
>
> As mentioned above, we will share the code and the atlas in a publicly available github repository in the final version of the paper.
>
> Regarding cintribution (2):
>
> The pipeline currently used to obtain the localization of the lesions performs registration from the subject space to the atlas MNI space (Montreal Neurological Institute's space - widely utilized in neuroimaging). The registration time is usually in the range of minutes per subject. Moreover, utilizing MNI space can introduce partial volume effects and distortion and increase structures' size in all directions. If there is a need to obtain subject's specific information about the regional lesion load, then it requires additional computations, considering introduced spatial changes.
>
> This is why we decided to try registering from the atlas space to the subject space. In theory, it should work similarly to the standard. However, in practice as mentioned in lines (146-149), the age-related changes and differences in the data resolutions can be too high to find proper transformation. Thus, the image intensities were not enough to register in this direction. Here, we found out that by extracting white matter, we're obtaining additional information, which is highly corrugated structures (instead of oval shapes) enough to register into the subject space and receive a ground truth mask in the subject space.
>
> With segmentation labels, we could train networks to segment the regions. Thus, the time necessary to obtain new labels equals the inference time, which is less than a minute per subject (typical approach: a few minutes per subject ). Ultimately, the details of the lesions can be calculated directly from the results obtained.
>
>
> 4.	The novelty of contribution (3) - "training" deep learning models cannot be considered a contribution by itself.
>
> The state-of-the-art models are all related to lesion segmentation. We haven't found in the literature any work directly related to segmenting WM regions for each subject by using deep learning. This is why this part is a novelty, and we opted for testing architectures known for their ability for accurate segmentation and different loss functions. As it is the first time WM regions are segmented using deep learning. It is possible because the masks in the subject's space are obtained before the process. Thus, considering region segmentation, our results are state-of-the-art and can be improved in future research.
>
> 5.	The limitations of the paper are not discussed.
>
> We add the limitation of our research in the discussion section. There are:
>
> •	The dataset used in this study does not contain a high resolution MRI data, some of them are almost 2D,
>
> •	The used dataset is not a longitudinal study, so our medical findings are based only on the available literature,
>
> 6.	In the Introduction, the previous and related work should be clearer, which can be exemplified by the following questions:Is this "the first method to fully automate WM region segmentation within a subject's anatomical space" (line 103)? Or is it the localization that is done for the first time in the native space?What are the disadvantages of working in the MNI space? Speed? Is the potential application of this method speed-dependent?
>
> •	It is exactly "the first method to fully automate WM region segmentation within a subject's anatomical space", as we mentioned in point 1.
>
> •	What are the disadvantages of working in the MNI space? Speed? - Please see answer to point 3.
>
> •	Is the potential application of this method speed-dependent? Yes, speed is one of the crucial factors when considering the location. By omiting registration to MNI space, we don’t introduce additional possible distortions. The processing time per subject is decreased from minutes to seconds which makes our solution robust and fast ennough for clinical research to move the focus from time-consuming and labor-intense approaches to using DL and concentrate on enhancing the field through medical findings.
>
> Thank you for your feedback; we hope that the public availability of the code will dispel your doubts and that we have addressed all other mentioned weaknesses and that they are now clear.

---

### Official Review · Reviewer_DDNP · 2024-10-09
**Lack of novelty but could be beneficial for audience with clinical background**

**Confidence:** 4

**Summary:**

The author developed several segmentation models for WMH lesion and region segmentation, with refind atlas labeled created and used. The propsoed framework could be used for WM lesion segmentation and localization on MICCAI 2017 WMH Segmentation Challenge dataset. Detailed analysis and discussions were conducted.

**Strengths:**

The author performed comprehensive experiments on WM lesion and region segmentation. Detailed discussion for results in relation to subject anatomy. The figures for the method and results are illustrative.

**Weaknesses:**

There are however lack of certain novelty opposing to what the author claimed for segmentation and localization. The discussion sometimes lack of depth. For example, when observing T1 is more significant for region/anotomical segmentation compared to that of FLAIR for lesion segmentation, the author could explore and explain such differences.

There are several possible technical errors and to be improved:
1. In Table 1, #scans for train and test were swapped.
2. What are the meanings behind various regions, as in Figure 3, the author directly mentioned region 10 and 12 are "larger regions
with simpler curvatures".
3. Why when reporting results, sometimes the author used CE+DS+SR loss, while in other places with CE+DS loss (e.g., in Appendix), especially the author concluded no significant differences between these loss functions (line 276-280).

**Justification:**

Although lack of novelty for method, dicusson and analysis could be a benneificial to readers (with clinical background).

---

> ### Author Rebuttal · Authors · 2024-10-22
>
> Reviewer DDNP,
>
> Thank you very much for noticing the valuable impact of our research on the medical field. We would like to clarify uncertain parts.
>
> 1.	Lack of certain novelty opposing to what the author claimed for segmentation and localization.
>
> As explained in lines (82-88) the existing studies are still lacking a fully-automatic and robust  solution that will follow the registration requirements and provide consistency in results.  This is why we introduce our method as the first one that fully automate WM region segmentation within subject’s anatomical space, eliminating the need for intra-subject or template-space registration, at the same time being fast enough for a clinical use (lines 102-113).
>
>
> In detail:
>
> The pipeline currently used to obtain the localization of the lesions performs registration from the subject space to the atlas MNI space (Montreal Neurological Institute's space - widely utilized in neuroimaging). The registration time is usually in the range of minutes per subject. Moreover, utilizing MNI space can introduce partial volume effects and distortion and increase structures' size in all directions. If there is a need to obtain subject's specific information about the regional lesion load, then it requires additional computations, considering introduced spatial changes.
>
> This is why we decided to try registering from the atlas space to the subject space. In theory, it should work similarly to the standard. However, in practice as mentioned in lines (146-149), the age-related changes and differences in the data resolutions can be too high to find proper transformation. Thus, the image intensities were not enough to register in this direction. Here, we found out that by extracting white matter, we're obtaining additional information, which is highly corrugated structures (instead of oval shapes) enough to register into the subject space and receive a ground truth mask in the subject space.
>
> With segmentation labels, we could train networks to segment the regions. Thus, the time necessary to obtain new labels equals the inference time, which is less than a minute per subject (typical approach: a few minutes per subject ). Ultimately, the details of the lesions can be calculated directly from the results obtained.
>
> Moreover, the majority of research relies on high-quality MRI data. As mentioned before, the known methods based on registration depend on the data quality. In practice, the MRI data used in-the-wild has lower quality. Thanks to training a network for region segmentation, we will no longer rely on the data quality, as we can obtain a generalized model that can produce accurate results regardless of data quality.
>
>
> 2.	For example, when observing T1 is more significant for region/anotomical segmentation compared to that of FLAIR for lesion segmentation, the author could explore and explain such differences.
>
> The T1 images, so-called structural images, provide good contrast between the tissues; therefore, the border of white matter is clearly separated from grey matter and cerebrospinal fluid. The number of characteristic curves on the borders is high, making the recognition of specific regions easier (see lines 227-229). In FLAIR imaging, the situation is the opposite. All healthy tissues have similar contrast, but the pathological parts are clearly separatable as they can be seen as a signal of increased intensity (see ;lines 271-274). Therefore, each modality with its unique properties is better suited for a different task, which was proved by the provided results.
>
> 3.	In Table 1, #scans for train and test were swapped.
>
> The #scans in the table were indeed swapped when comparing the original MICCAI2017 challenge, but they are correct when considering our research. Our paper aimed to highlight the importance of localization and the information it brings to the medical field instead of lesion segmentation and comparision with the WMH segmentation state-of-the-art. Thus, we swapped the original training and testing set to obtain more training data and focus on the paper's primary goal. As it remains unclear, we introduced this information into the 3.1 subsection.
>
> 4.	What are the meanings behind various regions, as in Figure 3, the author directly mentioned region 10 and 12 are "larger regions with simpler curvatures".
>
> The regions are provided in details in Tables A1 and A2 in Appendix. We refer to the numerical labels to save the space in manuscript. Region 10 contain Superior, Inferior and Middle Teporal Gyruses while region 12 contain Superior Corona Radiata and Superior Longitudinal Fasciculus, which are now directly mentioned in paper.
>
> 5.	Why when reporting results, sometimes the author used CE+DS+SR loss, while in other places with CE+DS loss (e.g., in Appendix), especially the author concluded no significant differences between these loss functions (line 276-280).
>
> Section 2.2 introduced three different loss functions we tested as a combination. The mentioned names "CE+DS+SR loss" and "CE+DS loss" stand behind their combination to directly refer which combination was used (see lines 168-172). Lines 276-280 refer to lesion segmentation, where no significant change in accuracy was observed while using all three functions. Nevertheless, in lines 236-240, we mentioned that the region segmentation model achieved the highest accuracy while training on combining all three losses.
>
>
> We hope we adress all unclear parts or technical issue mentioned in your feedback. If there is anything remaining or You have other questions, please let us know. Thank You.

---

### Official Review · Reviewer_1gxr · 2024-10-09
**Some unclarity regarding the method design and conclusions from the experiments**

**Confidence:** 3

**Summary:**

The proposed method performs simultaneous segmentation of white matter lesions and anatomical white matter regions. Some essential aspects of the methodology remained unclear to me. In general the manuscript is well written and understandable, but it could benefit from rewriting some part on why and how the loss terms are combined and what the benefit is of combining the localization and segmentation for the downstream task (of localization?). The results in the appendix show that many experiments have been performed, but some more work is needed to guide the reader through the experiments and their conclusions.

**Strengths:**

-	Relevant biomedical application
-	Dataset publicly availabe
-	Dataset from three different hospitals
-	Dataset covering multiple vendors
-	Many experiment results in appendix

**Weaknesses:**

-	It would have been interesting to see generalization capabilities by training on only data from e.g. Utrecht and Singapore and test on Amsterdam or similar.
-	Limited novelty regarding the methodology. Why is the segmentation needed if there are ways to automatically produce the ground truth?
-	Double check to introduce the full name before using the abbreviation (e.g. MRI, MNI,..)
-	The manuscript is generally clearly and well written, but there are some things that are unclear regarding the method. How is the ground truth for the WM region segmentation generated? Referring to Fig. 1A was not enough to understand the method. Why is a network trained to predict it if there is already a method to produce it?
-	What are the “load of WM lesions”? Please define/introduce.
-	The tables in appendix are not addressed in text. While they show the extend of the study, they should be put in context and discussed in the text.
-	How are the loss components weighted?
-	How to choose k in the k-means clustering step?

**Final Rebuttal Confidence:**

3

**Final Rebuttal Justification:**

The authors addressed many of the points that I found unclear in the initial submission. There may be a lack of novelty regarding the methodology in segmentation, as also pointed out by reviewer DDNP, but I do agree with both reviewers on the works' potential relevance of to the application of localization and segmentation of white matter lesions.

**Justification:**

Some more clarity is needed to understand why the tasks combined in the loss are relevant and provide an advantage of doing only the eventual downstream task. Some more explanations about the experiments reported in the appendix is needed to understand the study's findings.

---

> ### Author Rebuttal · Authors · 2024-10-22
>
> Dear Reviewer 1gxr,
>
>
> Thank you very much for your comments and questions. We believe that addressing them will benefit our research and clarify certain parts. We tried summarising and answering your points to rectify and explain your concerns.
>
> 1. It would have been interesting to see generalization capabilities by training on only data from e.g. Utrecht and Singapore and test on Amsterdam or similar.
>
> That is indeed a good point that we are currently investigating not only for this public dataset but also for a big inhouse dataset from different cohorts.
>
>
> 2.	Why is the segmentation needed if there are ways to produce the ground truth automatically?
>
> As explained in lines (82-88) the existing studies are still lacking a fully-automatic and robust  solution that will follow the registration requirements and provide consistency in results.  This is why we introduce our method as the first one that fully automate WM region segmentation within subject’s anatomical space, eliminating the need for intra-subject or template-space registration, at the same time being fast enough for a clinical use (lines 102-113).
>
>
> In detail:
>
> The pipeline currently used to obtain the localization of the lesions performs registration from the subject space to the atlas MNI space (Montreal Neurological Institute's space - widely utilized in neuroimaging). The registration time is usually in the range of minutes per subject. Moreover, utilizing MNI space can introduce partial volume effects and distortion and increase structures' size in all directions. If there is a need to obtain subject's specific information about the regional lesion load, then it requires additional computations, considering introduced spatial changes.
> This is why we decided to try registering from the atlas space to the subject space. In theory, it should work similarly to the standard. However, in practice as mentioned in lines (146-149), the age-related changes and differences in the data resolutions can be too high to find proper transformation. Thus, the image intensities were not enough to register in this direction. Here, we found out that by extracting white matter, we're obtaining additional information, which is highly corrugated structures (instead of oval shapes) enough to register into the subject space and receive a ground truth mask in the subject space.
> With segmentation labels, we could train networks to segment the regions. Thus, the time necessary to obtain new labels equals the inference time, which is less than a minute per subject (typical approach: a few minutes per subject ). Ultimately, the details of the lesions can be calculated directly from the results obtained.
>
> 3.	Double check to introduce the full name before using the abbreviation (e.g. MRI, MNI,..)
>
> We corrected omitted abbreviations and made sure all are now fully explained.
>
> 4.	How is the ground truth for the WM region segmentation generated?
>
> This question is related to Figure 1A, now updated to dispel doubts about the process and with some additional visual details mentioned by reviewer kk7q.
>
> The process of generating GT labels for region segmentation consists of the following steps:
>
> 1.	The white matter is extracted separately from subject T1 data and atlas T1.
>
> 2. Then the registration of the extracted atlas's WM to the subject's WM is performed to obtain affine transformation (Update 1 to the image: The "registration" block is moved up to the level of "subject T1" and renamed as "Registration to subject’s space" to indicate to which space we are registering.
>
> 3. The parallel process (done only once) is refining the atlas. The original atlas contains 130 labels and is too detailed; thus, it is refined into 34 regions, similar to other existing studies. (Update 2 to the image: We added the 'original atlas" image below "Atlas T1" that is pointing into Atlas labels refined, also moved to the bottom of block 1A.)
>
> 4. The last step is to apply the transformation obtained in step 2 to the 'atlas labels refined", which transforms them into subject space, providing subjects' ground truth labels. (Update 3 to the image: blocks from points 2 and 3 are combined, pointing out the final result - "Registered atlas labels").
>
> We hope the process, especially the image, will be straightforward and understandable.
>
>
> 5. Why is a network trained to predict it if there is already a method to produce it?
>
> The majority of research relies on high-quality MRI data. As mentioned before, the known methods based on registration depend on the data quality. In practice, the MRI data used in-the-wild has lower quality. Thanks to training a network for region segmentation, we will no longer rely on the data quality, as we can obtain a generalized model that can produce accurate results regardless of data quality.
>
> 6.	What are the "load of WM lesions"?
>
> We now defined it in line 31 of introduction section as “ The total WMH load, which is the volume of WM lesion in mm^3, increases with age and is…” .
>
>
> 7. The tables in appendix are not addressed in text. While they show the extend of the study, they should be put in context and discussed in the text.
>
> We believe all tables placed in appendix were adressed in the text:
>
> -	Lines 139-140 for Tables A1 and A2
>
> -	Lines 235-236 for Tables B1 to B5
>
> -	Lines 281-182 for Tables C1 to C5
>
> -	Line 299 for Table D1
>
>
> 8.	How are the loss components weighted?
>
> As the losses alone were almost in the same range, we decided to weight them equally. We added this information to second paragraph in section 3.2 (Settings).
>
> 9.	How do we choose k in the k-means clustering step?
>
> We selected the k based on silhouette index, with the best values obtained for k=3 and k = 4. We experimented with k = 4, to inspect the lesion load in different locations with more detailed characteristics.
>
>
> We would like to thank you for the valuable feedback that indeed allowed us to enhance the quality of our research. If anything remains unclear, please let us know; we will gladly explain and improve it.

---

### Meta-Review · Area_Chair_8o9F · 2024-10-25

**Recommendation:** Accept (Oral)
**Confidence:** 5

**Metareview:**

The paper has potential novelty and the authors answered the reviewers' comments.

**Suggested Changes To The Recommendation:**

3: I agree that the recommendation could be moved up

---

### Meta-Review · Area_Chair_1y1g · 2024-11-05

**Recommendation:** Accept (Poster)
**Confidence:** 5

**Metareview:**

The authors present an automated deep learning method for segmenting and localising white matter lesions (WMH) in neurological diseases.
The methodology involved using state-of-the-art deep learning models, such as U-Net, UNETR, MultiResUNet, and MedNeXt, to segment WMH and anatomical white matter regions simultaneously. The study is based on the MICCAI 2017 WMH Segmentation Challenge dataset, which includes 3D T1 and FLAIR images from 170 subjects across three cohorts.
The results showed that deep learning-based methods achieved high accuracy in WM lesion segmentation and localisation, with FLAIR images providing a clearer distinction between WM lesion and tissue intensities. The study also demonstrated the importance of location-specific analysis, highlighting the significance of WMH location in disease risk.
Overall the reviewers were positive in terms of the writing and clarity of the paper. Some concerns around novelty were well rebutted, given the paper is primarily focusing on the application of deep learning towards creating an automated system for WMH segmentation rather than a new deep learning approach.
My only concern is that the authors proposed some changes to the paper, e.g. changing to Figure 1 and providing the code and Atlas. Therefore, there is an expectation that all these changes will be added to the camera ready version and will be verified by the AC and/or PCs.

**Suggested Changes To The Recommendation:**

3: I agree that the recommendation could be moved up

---

### Decision · Program_Chairs · 2024-11-06

**Decision:**

Accept (Oral)

**Comment:**

We recommend an oral and a poster presentation given the AC and reviewers recommendations.